# Chiral inversion induced by aromatic interactions in short peptide assembly

Kai Qi [1], Hao Qi[1], Muhan Wang[2], Xiaoyue Ma[1], Yan Wang[1], Qiang Yao[1], Wenliang Liu[1], Yurong Zhao[1], Jiqian Wang [1], Yuefei Wang [3] ✉, Wei Qi [3], Jun Zhang[4] ✉, Jian R. Lu [5] ✉ & Hai Xu [1] ✉

Although hydrophobic interactions provide the main driving force for initial peptide aggregation, their role in regulating suprastructure handedness of higher-order architectures remains largely unknown. We here interrogate the effects of hydrophobic amino acids on handedness at various assembly stages of peptide amphiphiles. Our studies reveal that relative to aliphatic side chains, aromatic side chains set the twisting directions of single β-strands due to their strong steric repulsion to the backbone, and upon packing into multi-stranded β-sheets, the side-chain aromatic interactions between strands form the aromatic ladders with a directional preference. This ordering not only leads to parallel β-sheet arrangements but also induces the chiral flipping over of single β-strands within a β-sheet. In contrast, the lack of orientational hydrophobic interactions in the assembly of aliphatic peptides implies no chiral inversion upon packing into β-sheets. This study opens an avenue to harness peptide aggregates with targeted handedness via aromatic side-chain interactions.

In biological systems, biomolecules often form ordered assemblies to execute their functions, and many of them adopt helical suprastructures at higher structural levels, such as DNA double helices, DNA supercoils, α-helices, collagen triple helices, and cell microtubules. Because the helical handedness of these biomolecular assemblies has a profound impact on their properties and functions (e.g., molecular recognition, gene transcription, enzymatic catalysis, signal transduction, cell motility, and biological optics), understanding the suprastructure handedness and harnessing the biomolecular assembly process for targeted chiral suprastructures are of fundamental significance in the scientific and engineering communities.

It has long been accepted that the suprastructure handedness of biomolecular assemblies is dependent on the chirality of their building blocks. Many studies have attempted to link molecular chirality to supramolecular handedness[1-5]. For example, because single β-strands have been generally perceived to twist in the right-handed direction

for natural peptides composed of L-amino acids, subsequent backbone-backbone hydrogen-bonding (H-bonding) would drive them to pack into left-handed β-sheets, followed by the lateral association of these β-sheets into left-handed morphologies. Many natural β-sheet peptide nanofibrils reported so far display left-handed helicity, consistent with this hypothesis. However, several studies have indicated the preference for right-handedness by some L-form peptides[6-10]. For example, SAA$_{1-12}$ (RSFFSFLGEAFD) from the N-terminal of the serum amyloid A protein and ILQINS from hen egg white lysozyme assembled into right-handed fibrils[6-8]. For another example, designed Fmoc-FF*y* and Fmoc-FW*y* peptides (Fmoc denotes 9-fluorenylmethoxycarbonyl and *y* denotes polar residues including positively charged histidine (H) and arginine (R), negatively charged glutamic acid (E) and aspartic acid (D), and uncharged serine (S)) self-assembled into left- and right-handed helical nanofibers, respectively[9]. These conflicting results must imply a rather complicated mechanism

[1]State Key Laboratory of Heavy Oil Processing and Department of Biological and Energy Chemical Engineering, China University of Petroleum (East China), 66 Changjiang West Road, Qingdao 266580, China. [2]Department of Civil Engineering, Qingdao University of Technology, Qingdao 266033, China. [3]State Key Laboratory of Chemical Engineering, School of Chemical Engineering and Technology, Tianjin University, Tianjin 300072, China. [4]School of Materials Science and Engineering, China University of Petroleum (East China), Qingdao 266580, China. [5]Biological Physics Group, Department of Physics and Astronomy, The University of Manchester, Manchester M13 9PL, United Kingdom. ✉e-mail: wangyuefei@tju.edu.cn; zhangjunupc@upc.edu.cn; j.lu@manchester.ac.uk; xuh@upc.edu.cn

that underlines the suprastructure handedness of biomolecular assemblies. Such a complexity primarily arises from the highly hierarchical processes of biomolecular self-assembly and multiple nonbonded interactions involved.

H-bonding, hydrophobic interactions, and ionic interactions are the main driving forces involved in biomolecular assembly. Because of its strong directionality, backbone-backbone H-bonding plays an important role in directing the helical packing of twisted β-strands[1–5]. Unlike peptide backbones that cross the whole assemblies, charged side chains that contribute to ionic interactions are commonly distributed on the outer surface of the assemblies. As a result, they can define the surface topology of biomolecular assemblies via attractive and repulsive interactions and somewhat affect their suprastructure handedness. Consistent with this presumption, the helical handedness of the nanofibrils formed by the stereoisomers of the short amphiphilic $I_3K$ was governed by the chirality of the C-terminal Lys (K) residue, irrespective of the chirality of the hydrophobic Ile (I) residues[11]. Stereoisomers $^{L}I_3{}^{L}K$, $^{La}I_3{}^{L}K$, and $^{D}I_3{}^{L}K$ formed left-handed nanofibrils while $^{D}I_3{}^{D}K$, $^{Da}I_3{}^{D}K$, and $^{L}I_3{}^{D}K$ self-assembled into right-handed nanofibrils ($^{Da}I$ and $^{La}I$ denote D- and L-allo-form isoleucine). In contrast, in order to minimize hydrophobic-water interactions and maximize the van der Waals bonding of the hydrophobic groups, hydrophobic side chains prefer to be buried in the interior of aggregates in aqueous solution. Because hydrophobic interactions are usually nondirectional[12,13], they generally produce little influence on the suprastructure handedness of biomolecular assemblies, despite their intimate coupling with other non-bonded interactions in promoting ordered assembly.

However, when aliphatic hydrophobic Ile residues of $I_3K$ were replaced with aromatic hydrophobic phenylalanine (Phe or F) ones, our study revealed that the helical handedness of the $F_3K$ fibrils was dictated by the chirality of Phe rather than Lys [14]. $^{L}F_3{}^{L}K$ and $^{L}F_3{}^{D}K$ self-assembled into left-handed β-sheet nanofibrils, while $^{D}F_3{}^{L}K$ and $^{D}F_3{}^{D}K$ formed right-handed ones. Such a result prompts us to reconsider the role of the hydrophobic interactions in controlling the supramolecular handedness. In fact, due to the geometric restrictions of the rigid aromatic rings, their continuous association can provide a certain degree of orientational preference, termed π-π stacking. On the other hand, when we reexamine the peptides that form unusual suprastructure handedness, we observe that most of them are rich in aromatic residues, such as RSFFSFLGEAFD and Fmoc-FFR[6,7,9]. Despite these cues and conjectures associated with aromatic side-chain interactions, how aromatic side chains regulate the suprastructure chirality of peptide assemblies and how such interactions differ from aliphatic side-chain interactions and combine with directional backbone-backbone H-bonding are still unclear.

In a recent study, we have developed a theoretical method combining quantum chemistry (QC) calculations and molecular dynamics (MD) simulations, which enables the reliable determination of single β-strand conformations and chirality based on intrastrand non-bonded interactions[15]. Simultaneously, although the designed short amphiphilic peptides, including aliphatic $I_3K$ and aromatic $F_3K$, Fmoc-FF*y*, and Fmoc-FW*y* are akin in terms of molecular amphiphilicity and peptide sequence and length, their β-sheet assemblies showed distinct suprastructure handedness[9,11,14]. These systems are, hence, ideal model peptides enabling the interrogation of aromatic interactions in regulating the suprastructural handedness, using aliphatic hydrophobic interactions as a control.

In light of the key features of these peptides, we here shift our interest from their single molecule conformations to their assembly processes beyond molecules, focusing on the interstrand interactions, including backbone-backbone H-bonding, side chain-side chain hydrophobic interactions, and their interplay, as well as interstrand arrangements. The main aim is to uncover the specific role of aromatic interactions in regulating the suprastructure handedness of β-sheet peptide nanofibrils. To attain this end, a variety of theoretical analyses are introduced to visually and quantitatively analyze the interstrand interactions at a higher level of assembly. We demonstrate that the formation of aromatic ladders between β-strands causes chiral flipping over of single strands upon packing into a β-sheet.

## Results

### Single β-strand conformations and handedness

Single β-strands tend to twist due to the innate chirality of natural amino acids. Using the established theoretical method[15], our structural optimization for single strands showed that the handedness of single $F_3K$ strands was governed by the chirality of the aromatic residues, i.e., single $^{L}F_3{}^{L}K$ and $^{L}F_3{}^{D}K$ β-strands were left-handed twisted while $^{D}F_3{}^{L}K$ and $^{D}F_3{}^{D}K$ ones were right-handed, irrespective of the charged state of Lys4 (Fig. 1a–d and Supplementary Fig. 1a–d). This is in sharp contrast to their aliphatic counterparts $I_3K$ strands, where the handedness was dictated by the chirality of the C-terminal Lys residue. For example, single $^{D}I_3{}^{L}K$ and $^{D}I_3{}^{D}K$ strands were right- and left-handed twisted, respectively (Fig. 1e, f and Supplementary Fig. 1e, f). In contrast, single $^{L}I_3{}^{D}K$ and $^{L}I_3{}^{L}K$ β-strands were left- and right-handed twisted[15]. It is evident that $^{D}F_3{}^{D}K$ and $^{D}I_3{}^{D}K$ or $^{L}F_3{}^{L}K$ and $^{L}I_3{}^{L}K$ strands exhibited the opposite handedness at this stage.

To reveal the molecular explanation for this conformational difference, we performed the reduced density gradient (RDG) isosurface analysis. Such a method allows simultaneous analysis and visualization of various noncovalent interactions such as van der Waals interactions, H-bonds, and steric repulsion as real-space surfaces for molecular complexes[16]. Within the RDG isosurfaces, the blue, green, and red areas represent strong attractive H-bonding, weak van der Waals interactions, and strong steric repulsion, respectively[16,17]. We found that strong steric hindrance between the benzene ring of Phe3 and the local backbone (between Phe3 and Lys4), as denoted by a larger brown patch in the enlarged area of Fig. 1g or Supplementary Fig. 1g, repelled the C=O and N-H groups of Lys4 to shift downward for $^{D}F_3{}^{D}K$ and $^{D}F_3{}^{L}K$ or upward for $^{L}F_3{}^{L}K$ and $^{L}F_3{}^{D}K$, irrespective of the chirality of Lys4 (Fig. 1g, h, and Supplementary Figs. 1g, h, 2). In contrast, there was a lack of such a strong steric hindrance between the isobutyl group of Ile3 and the local backbone for $I_3K$, and instead, the orientation of the C=O and N-H groups of Lys4 was dictated by its long side chain (Fig. 1i and Supplementary Figs. 1i, 2).

### Handedness inversion induced by aromatic interactions in β-sheet assembly

Based on the above single-strand conformations, right-handed β-sheets were expected for $^{L}F_3{}^{L}K$ and $^{L}F_3{}^{D}K$ assembly, while left-handed ones for $^{D}F_3{}^{L}K$ and $^{D}F_3{}^{D}K$. However, left-handed β-sheet nanofibrils were observed to be formed by the former two while right-handed twisted ones by the latter two in our experiments[14]. This is distinct from the self-assembly of $I_3K$ peptides, in which right- and left-handed β-strands eventually packed into left- and right-handed β-sheet nanofibrils[11,15].

To determine the mechanism underpinning such a discrepancy, we focused on β-strand dimers. As a fundamental structural motif of β-sheets and β-sheet assembly, dimers enable us to analyze interstrand interactions such as backbone H-bonding and side-chain interactions. The independent gradient model-based Hirshfeld partition (IGMH, 0.005 a.u.) was adopted to graphically describe these weak interactions[17,18]. Dimers were first constructed based on the single-strand conformations described in Fig. 1 or Supplementary Fig. 1, via parallel and anti-parallel arrangements. After structural optimization, the IGMH analysis indicated extensive interstrand π−π interactions within the most stable $F_3K$ dimers, indicated by the green regions between benzene rings, for the three interstrand contacting modes, namely FF (face-to-face), FB (face-to-back), and BB (back-to-back), as

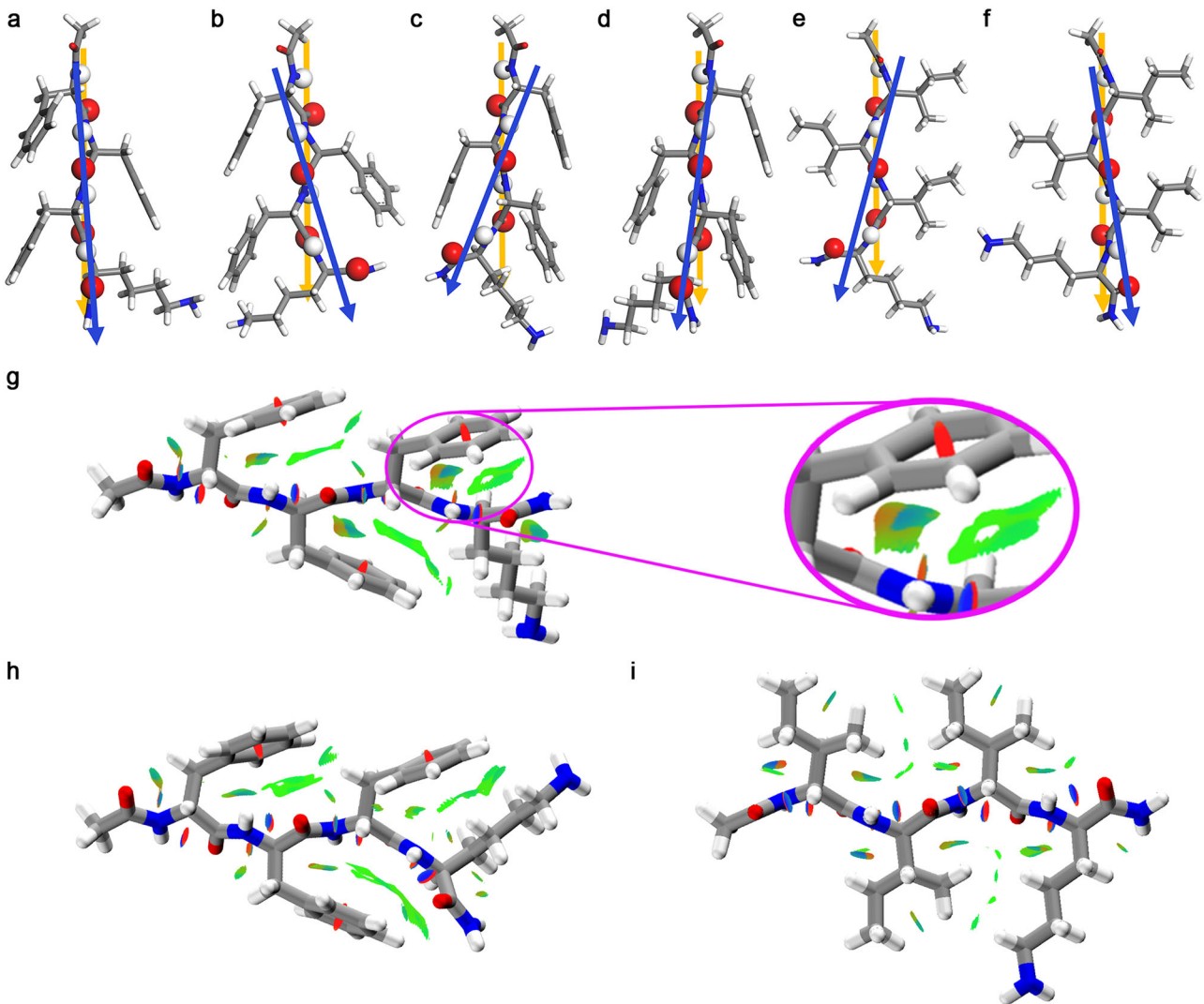

**Fig. 1 | Single-strand conformations, handedness, and intrastrand interactions.**
**a–f** Twisting directions of single β-strands for (**a**) $^{L}F_3{}^{L}K$, (**b**) $^{L}F_3{}^{D}K$, (**c**) $^{D}F_3{}^{L}K$, (**d**) $^{D}F_3{}^{D}K$, (**e**) $^{D}I_3{}^{L}K$, and (**f**) $^{D}I_3{}^{D}K$. **g–i** RDG isosurfaces of single $^{D}F_3{}^{D}K$, $^{D}F_3{}^{L}K$, and $^{D}I_3{}^{D}K$ β-strands, respectively. Atoms coloring scheme is: red, oxygen; blue, nitrogen; white, hydrogen, and gray, carbon. The two arrowed lines (from N- to C-terminus: blue and yellow) connecting potential H-bonding atoms on each side can define the twisting directions of the single strands. The blue, green, and red areas in the RDG isosurfaces represent attractive H-bonding, van der Waals interactions, and steric repulsion, respectively. When these peptides were positively charged via the side-chain protonation of Lys4, they showed the same twisting directions (Supplementary Fig. 1). Gradient isosurfaces of the other three peptide β-strands are given in Supplementary Fig. 2.

shown in Fig. 2 and Supplementary Fig. 3 for $^{L}F_3{}^{L}K$ and Supplementary Figs. 4, 5 for $^{D}F_3{}^{L}K$. The three modes are schematically depicted in Supplementary Fig. 6 for parallel and anti-parallel arrangements. Furthermore, interstrand π-π interactions were found to be stronger within the parallel dimers (left halves of Fig. 2 and Supplementary Figs. 3–5), as indicated by the larger green areas, compared to the anti-parallel dimers (right halves of Fig. 2 and Supplementary Figs. 3–5). For the FB contacting mode within the parallel dimer, there were three pairs of side-chain π-π stacking while only two pairs for the same mode within the anti-parallel dimer. Additionally, the side-chain π-π stacking for the parallel arrangements was parallel-displaced while T-shaped for the anti-parallel arrangements. These IGMH results suggested a high propensity of $F_3K$ β-strands for parallel arrangements upon packing, presumably driven by stronger side-chain π-π stacking interactions.

To further determine which arrangement was preferable, we calculated the binding energies of the $F_3K$ dimers. In contrast, the $^{D}I_3{}^{L}K$ and $^{D}I_3{}^{D}K$ dimers were also included. Considering the three FB, FF, and BB contacting modes and the charged state of Lys4, 72 optimized dimer conformations were employed for such a calculation (Supplementary Table 1). For aliphatic $I_3K$ peptides, the dimers with anti-parallel arrangements always exhibited lower binding energies than the parallel ones, irrespective of their contacting modes (FF, FB, and BB) and charged states (neutral and positively charged). For aromatic $F_3K$ peptides, however, the dimers with parallel arrangements always displayed lower binding energies than the anti-parallel ones, even for the charged ones. Due to the possible electrostatic repulsive interactions between positively charged Lys4 side chains, the difference in the binding energy for $F_3K$ between the parallel and anti-parallel arrangements mostly decreased when the dimers were positively charged through the protonation of Lys4 side chains.

Although it is generally perceived that the backbone H-bonding is slightly stronger for anti-parallel arrangements than parallel ones, we here demonstrated stronger parallel H-bonding interactions within $F_3K$ dimers, by using the core-valence bifurcation (CVB) index (refer to Supplementary Method 1 for the CVB index definition). Through the topological analysis of the electron localization function, the index can quantitively describe the H-bonding strength: the lower the CVB index

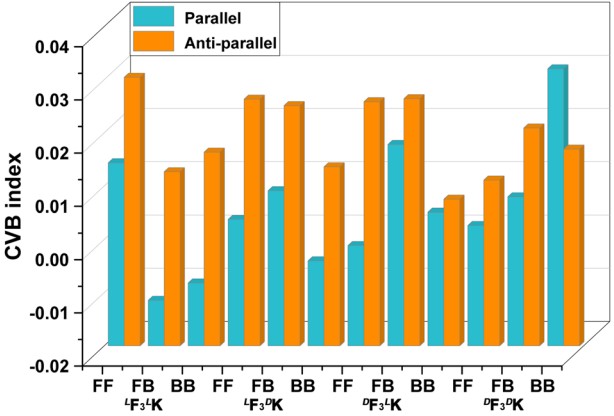

**Fig. 2 | Dimer conformations, π-π stacking interactions between β-strands, and chiral inversion.** IGMH analysis of π-π stacking interactions between β-strands within parallel and anti-parallel $^{L}F_3{}^{L}K$ dimers. These dimers were obtained after structural optimization, based on the single-strand conformations given in Fig. 1. Compared to the anti-parallel dimers, more extensive and stronger π-π stacking interactions were revealed within the parallel ones, as denoted by green patches in the IGMH isosurfaces. Importantly, chiral inversion of single strands only happened within the parallel dimers, irrespective of the contact modes between strands. Atoms coloring scheme is: red, oxygen; blue, nitrogen; white, hydrogen, and gray, carbon. The IGMH analysis for charged $^{L}F_3{}^{L}K$ dimers is given in Supplementary Fig. 3.

**Fig. 3 | CVB indices of $F_3K$ β-sheet dimers.** Each dimer has six conformations that are differentiated by parallel and anti-parallel arrangements and three contacting modes. For each dimer conformation, the index shown was the average value of the indices of all interstrand H-bonds.

is, the stronger the H-bond will be[19]. Except for the BB mode of $^{D}F_3{}^{D}K$, all the parallel $F_3K$ dimers exhibited lower CVB indices after structural optimization than the anti-parallel ones for each contacting mode (Fig. 3). Such a result was virtually consistent with previous predictions for parallel β-sheets favorably formed by aromatic Phe residues[20]. We further verified that the parallel-displaced orientation of benzene rings within parallel arrangements led to the formation of aromatic ladders (Fig. 4, as discussed below). Such a well-ordered packing mode was most likely to greatly alleviate the potential intermolecular steric hindrance, thereby constituting the main drive for the formation of stronger H-bonding interaction between parallel backbones. In contrast, there was a lack of aromatic ladders for anti-parallel arrangements (Supplementary Fig. 7).

The most striking finding with the dimers was that chiral inversion occurred for single β-strands for all the parallel arrangements (FF, FB, and BB). For example, $^{L}F_3{}^{L}K$ β-strands turned into right-handed ones (left halves of Fig. 2 and Supplementary Fig. 3) and $^{D}F_3{}^{L}K$ β-strands converted into left-handed ones (left halves of Supplementary Figs. 4, 5). In contrast, there was no chiral inversion within all the anti-parallel dimers. For

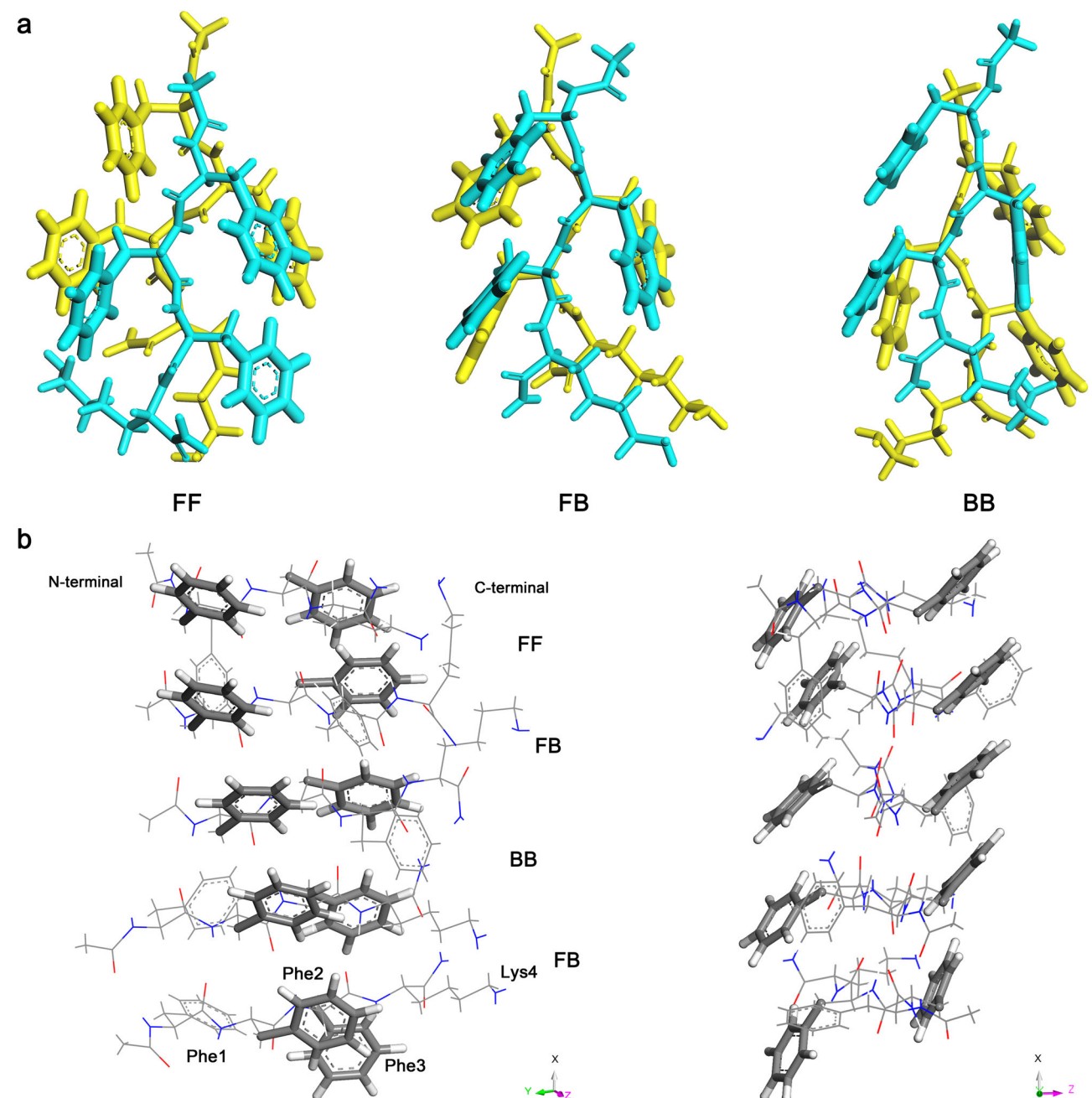

**Fig. 4 | Parallel $^{L}$F$_{3}$$^{t}$K dimers and pentamer. a** Parallel $^{L}$F$_{3}$$^{t}$K dimers with FF, FB, and BB contacting modes. Yellow and blue colors are used to distinguish the two molecules. **b** Parallel $^{L}$F$_{3}$$^{t}$K pentamer containing one FF, one BB, and two FB modes, in which two side-chain aromatic ladders were formed along the long axis of the β-sheet. Atoms coloring scheme is: red, oxygen; blue, nitrogen; white, hydrogen, and gray, carbon.

example, $^{L}$F$_{3}$$^{t}$K and $^{D}$F$_{3}$$^{t}$K β-strands adopting anti-parallel arrangements remained left- and right-handedness, respectively (right halves of Fig. 2 and Supplementary Figs. 3–5). This chiral inversion within the parallel dimers was indeed driven by the parallel-displaced contacts between side-chain benzene rings (Fig. 4a). Such a contacting mode of benzene rings not only caused stronger π-π stacking but also strengthened H-bonding between backbones via decreasing intermolecular steric hindrance, as described above. As a result, the parallel dimers with chiral inversion exhibited lower binding energies (Supplementary Table 1) and were thus thermodynamically favored, compared to the anti-parallel ones in which the strand handedness remained unchanged (Supplementary Fig. 7a).

Based on the optimized dimers (parallel, Fig. 4a; anti-parallel, Supplementary Fig. 7a), the Kabsch algorithm was then applied to construct assemblies at higher structural levels along the H-bonding direction. This method enables the best alignment of β-strands within their assemblies, by calculating the optimal rotation matrix that minimizes the root mean squared deviation (RMSD) between two paired sets of points (backbone H-bonding donors and acceptors)[21]. Pentamers featuring a repeat of one FF, one BB, and two FB modes were constructed (Supplementary Fig. 8). After structural optimization, two aromatic ladders were clearly formed with the parallel pentamer along the H-bonding direction (Fig. 4b) while there was no such an ordered aromatic ladder for the anti-parallel pentamer (Supplementary Fig. 7b).

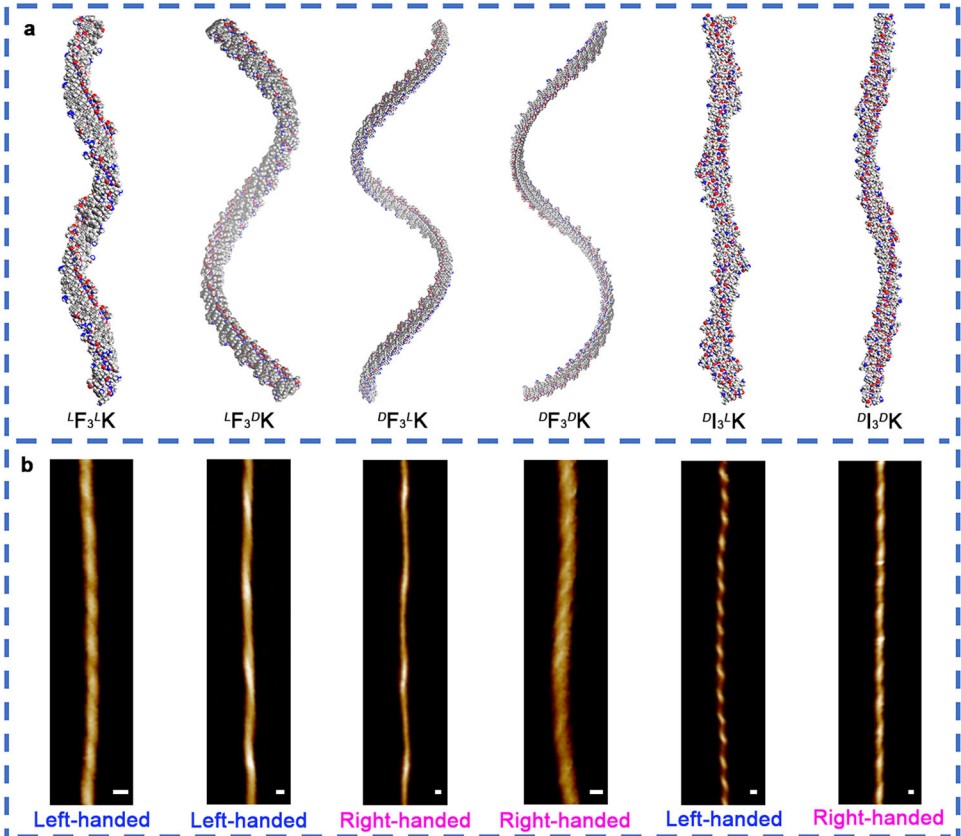

**Fig. 5 | Handedness of multi-stranded β-sheets and nanofibrils. a** Multi-stranded parallel β-sheets for $^LF_3^LK$, $^LF_3^DK$, $^DF_3^LK$, and $^DF_3^DK$ and anti-parallel β-sheets for $^DI_3^LK$ and $^DI_3^DK$ after structural optimizations. Atoms coloring scheme is: red, oxygen; blue, nitrogen; white, hydrogen, and gray, carbon. **b** Nanofibrils formed by the six peptides and their handedness. Scale bars, 20 nm.

These aromatic ladders between β-strands were reminiscent of the Phe zipper interaction between β-sheets, which was formed via their facial complementarity[22]. Further packing of the optimized pentamers along the H-bonding direction via the Kabsch algorithm gave rise to long β-sheets. After structural optimization using a semiempirical QC method, GFN0-xTB[23], parallel β-sheets exhibited left-twisted spirals for $^LF_3^LK$ and $^LF_3^DK$ and right-twisted spirals for $^DF_3^LK$ and $^DF_3^DK$ (Fig. 5a). At much higher structural level, lateral stacking of these left- and right-handed β-sheets will inevitably result in the formation of fibrils with left- and right-handedness, respectively, in good line with our experimental observations (Fig. 5b). Because no chiral inversion occurred for single strands within anti-parallel arrangements, anti-parallel β-sheets would exhibit right-twisted spirals for $^LF_3^LK$ and $^LF_3^DK$ and left-twisted spirals for $^DF_3^LK$ and $^DF_3^DK$, thus contradicting our experimental observations and excluding the possibility of these microscopic structures.

In contrast, there was no chiral inversion for I3K strands when packing into dimers, regardless of parallel and anti-parallel arrangements (Supplementary Fig. 9). The underlying mechanism was due to the lack of side-chain aromatic rings in I3K peptides, although Ile has similar hydrophobicity and β-sheet forming propensity to Phe[24,25]. As a consequence, the hydrophobic collapse of Ile side chains cannot provide a high orientation and a highly ordered side-chain packing, thus being unable to reverse the handedness of single strands upon packing into dimers and higher-order structures such as β-sheets and nanofibrils.

## Handedness evolution with other amphiphilic aromatic peptides

We extended our investigation on all L-form Fmoc-FF*y* and Fmoc-FW*y* peptides[9]. Although the right-handedness of Fmoc-FW*y* nanofibers

was proposed to be mainly caused by the intermolecular steric hindrance between bulky Trp side chains, the relationship between suprastructure handedness and aromatic interactions remains elusive.

Consistent with $^LF_3^LK$, our simulations indicated that single strands of Fmoc-FF*y* and Fmoc-FW*y* twisted in a left-handed direction (Fig. 6a, b, with Fmoc-FFR and Fmoc-FWR as the examples). When packing into β-sheet structures, these strands tended to adopt parallel arrangements within a β-sheet, imposed by the π-π inter-locking of Fmoc groups between adjacent β-sheets[26,27]. Upon forming dimers, chiral inversion happened for single β-strands within the optimized Fmoc-FFR dimeric structure (from left-handedness to right-handedness), in which the side-chain benzene rings adopted a parallel-displaced orientation and thus resulted in strong π-π stacking interactions (Fig. 6c), similar to $^LF_3^LK$ and $^LF_3^DK$. In our simulations, we deliberately constructed parallel Fmoc-FFR dimers without chiral inversion for single strands (Fig. 6d). The dimeric structure formed at the initial stage of structural optimization exhibited weaker side-chain π-π stacking interactions. Afterward, this structure rapidly converted into the structure shown in Fig. 6c, accompanied by the chiral inversion of single strands, again exhibiting a thermodynamically favorable chiral inversion. As expected, the optimized dimers containing right-handed strands further evolved into left-handed β-sheets (Fig. 6g), as demonstrated by the Kabsch algorithm, and these left-handed β-sheets would eventually associate into left-handed helical nanofibers, consistent with experimental observations (Fig. 6i).

In contrast, Fmoc-FWR β-strands remained left-handedness within the optimized dimeric structure (Fig. 6f), that is, no chiral inversion. More importantly, both benzene and indole rings were parallel-displaced in such a dimeric conformation, and there were strong π-π stacking interactions between the two strands, as denoted by the

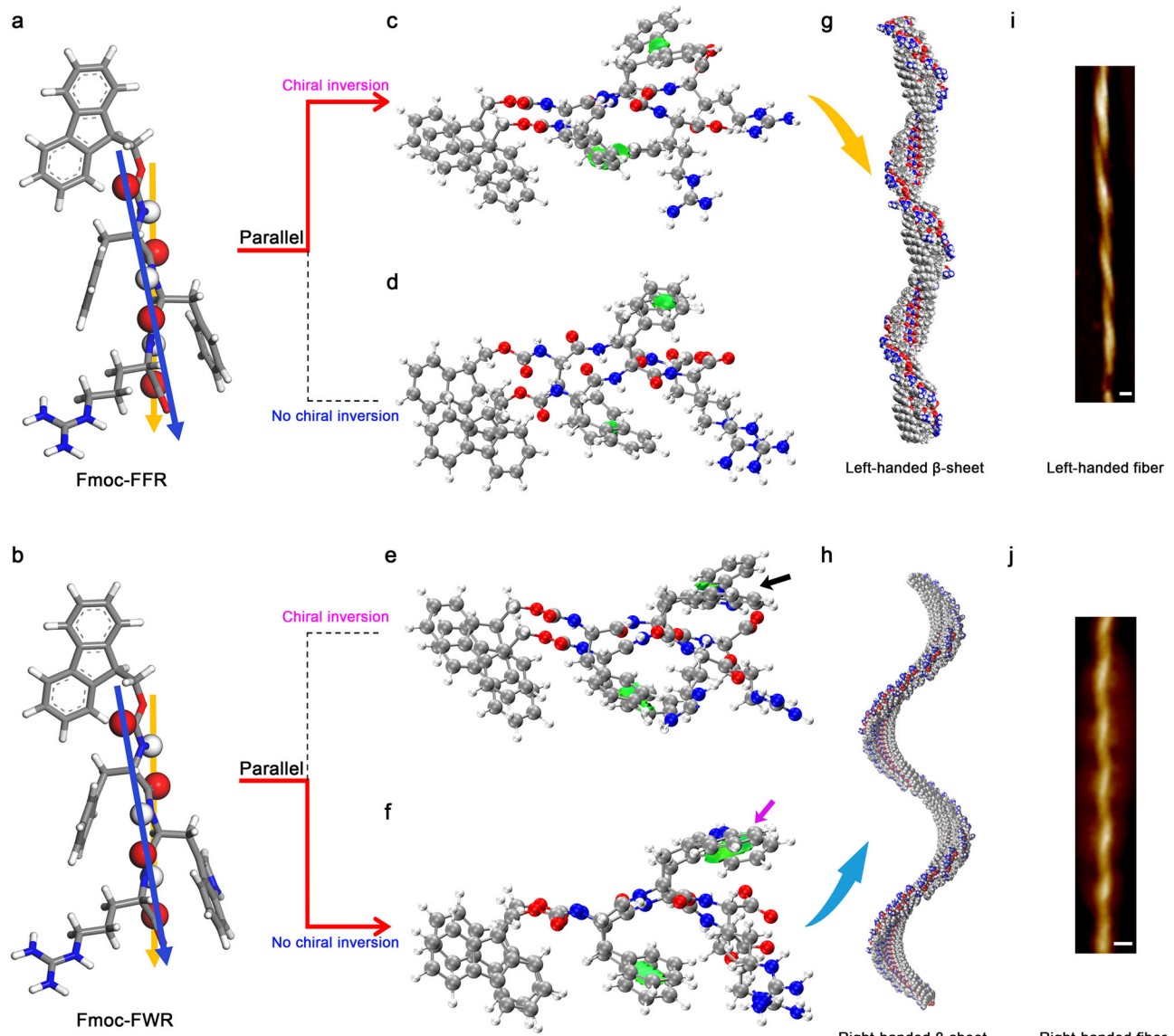

**Fig. 6 | Handedness evolution for Fmoc-FFR and Fmoc-FWR. a**, **b** Single β-strands. **c**–**f** Dimers. **g**, **h** Multi-stranded β-sheets. **i**, **j** Self-assembled β-sheet nanofibrils observed from experiments. Note that chiral inversion for single Fmoc-FFR strands occurred upon dimerization and further assembly, while it was not the case for Fmoc-FWR. Atoms coloring scheme is: red, oxygen; blue, nitrogen; white, hydrogen, and gray, carbon. Scale bars, 20 nm.

magenta arrow. The subsequent packing of the dimers along the H-bonding direction led to the formation of right-handed β-sheets (Fig. 6h), which would further associate into right-handed helical nanofibers (Fig. 6j). Similarly, we also constructed parallel Fmoc-FWR dimers with chiral inversion for single strands (from left-handed to right-handed). At the initial stage of our structural optimization, however, there was marked steric repulsion between two indole rings, making them less contact within such a structure (as indicated by the black arrow in Fig. 6e). At the final stage of structural optimization, the dimeric β-sheet structure collapsed.

## Discussion

The theoretical approach adopted in this study by combining QC calculations and MD simulations not only allowed the determination of single β-strand conformations and chirality based on intrastrand non-bonded interactions but also depicted how the handedness evolved upon assembly into multiple-stranded β-sheets based on interstrand interactions. The use of structurally similar aliphatic $I_3K$ and aromatic $F_3K$ peptide amphiphiles and their multiple stereoisomers enabled a straightforward and in-depth comparison of common aliphatic hydrophobic interactions and aromatic hydrophobic interactions. These analyses demonstrated the crucial roles of aromatic side chains in controlling molecular and supramolecular handedness. Specifically, their strong steric repulsion to the backbone led to the molecular chirality of $F_3K$ peptide stereoisomers distinctly different from that of $I_3K$ ones. Beyond the molecular level, the formation of aromatic ladders between strands could provide a directional preference for their ordered packing into multiple-stranded β-sheets, not only resulting in parallel arrangements but also inducing the chiral flipping over of single $F_3K$ β-strands. Thus, it was the directional aromatic interactions rather than backbone-backbone H-bonding that directed the handedness transition during the packing of $F_3K$ β-strands into β-sheets. On the contrary, there was a lack of the orientational hydrophobic interactions in aliphatic $I_3K$ assembly and thus no chiral inversion for $I_3K$ β-strands upon their packing into β-sheets. It should, however, be the directional backbone-backbone H-bonding that directed the handedness evolution during the packing of $I_3K$ β-strands into β-sheets.

When we extended our theoretical analysis on the all L-form Fmoc-FFR and Fmoc-FWR, we found that during the packing of β-strands into multiple-stranded β-sheets, the formation of interstrand ordered aromatic interactions favored the chiral inversion for single Fmoc-FFR β-strands but was in disfavor for the chiral inversion for single Fmoc-FWR β-strands, thus leading to β-sheet nanofibrils with the opposite handedness.

These results clearly reveal that the chirality selection within the self-assembly of the short amphiphilic peptides must be thermo-dynamically favored. This study not only provides an excellent expla-nation for understanding the roles of different hydrophobic interactions in controlling the handedness of peptide β-sheet assemblies, but also helps design unusual suprastructure handedness via the formation of ordered side-chain aromatic interactions in peptide assembly.

## Methods

### Simulations

Density functional theory (DFT) simulations were performed using Gaussian 09 (Rev. D. 01) software[28]. During the whole DFT simulation process, the generalized gradient approximation (GGA) scheme and the PBE0 exchange-correlation function were used[29]. The def2-SVP basis set (2-ζ base group) was used to optimize the molecular struc-ture, and the def2-TZVP basis set (3-ζ base group) was applied to cal-culate the conformational energy[30]. The DFT-D3 dispersion correction was employed to more accurately describe dispersion interactions[31].

First, a large number of molecular conformations were obtained by performing MD simulations on a single peptide molecule (see Supplementary Method 2 for MD simulation details). Based on the degree of extending and dihedral angles ($\varphi$ and $\psi$), 50 extended con-formations were selected for subsequent QC calculations. Through structural optimization and energy calculation, the single β-strand conformations with the lowest energies were obtained.

Then, the Molclus program was used to generate 200 dimers for FF, FB, and BB modes, respectively, and subsequently to search for stable dimeric structures[32]. Specifically, the 200 dimers were opti-mized using a semiempirical QC method, PM7[33], resulting in five dimeric configurations with the lowest energies. These stable dimer configurations were further optimized by DFT simulations. The counterpoise method developed by Boys and Bernardi was applied to correct the Basis Set Superposition Error (BSSE) in the process of dimer geometry optimization[34]. All charged systems were calculated using the solvation model based on density (SMD) solvation model with $H_2O$ as the solvent to simulate the solution environment[35]. The pentamers constructed by the Kabsch algorithm were also subjected to structural optimization using DFT simulations. After these simula-tions, all the snapshots were displayed using the VMD software[36].

### Self-assembled morphology and handedness characterizations

Peptide solutions were prepared by directly dissolving peptide solid powders in water and their solution pH values were adjusted to pH 6.0 using dilute HCl or NaOH. The peptide concentration was 8 mM for $F_3K$ and $I_3K$ and 2 mM for Fmoc-FFR and Fmoc-FWR, respectively. After incubation for 1 week at room temperature, their self-assembled nanostructures and suprastructure handedness were determined using atomic force microscopy (AFM) on a Bruker MultiMode 8 scan-ning probe microscope equipped with a NanoScope V controller under ambient conditions. Height, amplitude, and phase images were con-currently captured using the ScanAsyst mode in air and are presented after the first-order line fit flattening to correct for piezo-derived dif-ferences between scan lines.

## Data availability

All the data supporting this study are available within the main text and its supplementary information. These data were also available from the corresponding authors upon request.

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

## Acknowledgements

We acknowledge funding by the National Natural Science Foundation of China (22172193 to H.X., 22072181 to J.W., and 22278306 to Y.W.). We also thank AstraZeneca (R122338) and BBSRC LINK with AstraZeneca (BB/S018492/1 and BB/S018492/2) for funding support for this work and innovate UK for the Knowledge Transfer Partnership fellowships under KTP12697, KTP10809 and KTP11592.

## Author contributions

K.Q., J.W., Y.W., J.R.L. and H.X. conceived the concept and designed the study. K.Q., M.W., Y.W., Q.Y. and J.Z. performed the QC calculations and MD simulations. H.Q., X.M., Y.W., W.L., Y.Z. and Y.W. conducted microscopic imaging. K.Q., Y.W., W.Q., J.Z., J.R.L. and H.X. analyzed the data and wrote the manuscript. All authors discussed the results and reviewed the manuscript.

## Competing interests

The authors declare no competing interests.
