## [Peer Review File · Nature Communications]

REVIEWER COMMENTS

Reviewer #2 (Remarks to the Author):

In this work, the authors investigated the chiral inversion for Phe-based tetrapeptides. It was found the pi-pi stacking of Phe-Phe interactions plays a determinant role in reversing the chirality of the peptides from single-chain strands to multi-chain fibers. This work provides very interesting observations and mechanisms of peptide assembly. The manuscript was well written. It could be further improved with more details about building the structures (pentamers, etc.) and data analyses.

Major revisions:

1. What is the “Reduced density gradient”? A brief description is required as well as the reference(s).
2. It seems that Fig. S1(a1-a6) are the same as those in Fig. 1. While protonated Lys was used in Fig. S1b, charge-neutral Lys was provided in Fig. 1b. At neutral solution or pH = 6, Lys is supposed to be fully charged. I see no reason why the charge neutral Lys was provided in the main text.
3. How was the “core-valence bifurcation (CVB) index” defined? A brief description is desired.
4. (Page 22) “All charged systems were calculated using the solvation model based on density (SMD) solvation model with H₂O as the solvent.” Does it mean that the charge-neutral system was done in a vacuum? How strong are the influences of the implicit hydration models?
5. What is the Kabsh algorithm? Any reference? The pentamer structures were constructed using the Kabsh algorithm and were subjected to DFT optimization. Please justify that these structures are at an energetically global minimum. Without free energy calculations, the structures are seemingly arbitrary.
6. I am a bit surprised that no micellar structures were formed from these peptides. Atomistic MD simulations are desired to justify the fiber morphology, instead of micelles.

Minor revisions:

1. In the MD simulations and QM calculations, how were the C- and N-terminals treated? Both are seemingly charge neutral in Fig. 1. Please confirm.
2. (Pag 5-6) $\{La\}_3^L K$ & $\{Da\}_3^D K$. Replace La with L, and Da with D.
3. (Page 7) “brown patch in the enlarged area of Fig. 1b1”. I suppose the authors are referring to the “green patch”. Please clarify and be specific.
4. Please provide the full name of “QC”.

Reviewer #3 (Remarks to the Author):

This manuscript uses a previously developed theoretical method combining quantum chemistry calculations and molecular dynamics simulations in order to reliably determine the single Beta-strand conformations and chirality based on intrastrand non-bonded interactions. Their computational results are compared with synthesis and atomic force microscopy results. By providing an effective approach for unraveling how hydrophobic interactions control the handedness evolution of peptide beta-sheet assemblies, the study opens an avenue to harness peptide aggregates with targeted handedness via aromatic side chain interactions.

This investigation is of interest to a wide audience and appears to be original. The methods used are satisfactory, although I would be curious if qualitative results hold with different functionals. I do think the presentation could be improved to make the presentation and discussion of their results clearer. As it stands now it is work for the reader to figure out when theoretical predictions are being made vs. when they are explicitly being compared to experiment. Perhaps a table when discussing the Kabasch algorithm predictions. With a little bit of clarification and discussion of chosen methods I would recommend this paper for publication.

“Chiral inversion...” Kai Qi et al, Nat Comm. submission

This manuscript outlines an important and well-implemented study of the impact of side-chain associations as an energetic constraint on supramolecular assembly chirality. Using simple model systems, they have developed models of assembly energetics observed in many protein-misfolding diseases that have been overlooked to this point.

Specific points:

- The representation of these higher-order structures is notable, particularly in Fig 4, and should be addressed.
- These assembly processes are nucleation-dependent processes raising questions of thermodynamic vs kinetic control. This control is hard to address and they nicely approach this selection in their computational analyses and should be highlighted in their claims.
- The authors do not address the issue of facial complementarity, so prevalent in cross β assemblies (*J. Am. Chem. Soc.*, 130, 9829-9835), which also may impact nucleation and supramolecular propagation. Why was this issue not considered given its critical role in supramolecular nucleation?
- A recent publication in this journal (<https://doi.org/10.1038/s41467-024-45019-2>. **Editor’s Selection**) highlights the impact of Phe residues on supramolecular chirality but does not adequately resolve the kinetic vs thermodynamic control or the structural models for assembly, but should be referenced.

Chiral inversion induced by aromatic interactions in short peptide assembly: insights from simulations and experiments

Reviewer #1:

This manuscript outlines an important and well-implemented study of the impact of side-chain associations as an energetic constraint on supramolecular assembly chirality. Using simple model systems, they have developed models of assembly energetics observed in many protein-misfolding diseases that have been overlooked to this point.

Reply:

We thank the Reviewer for this positive assessment.

The representation of these higher-order structures is notable, particularly in Fig 4, and should be addressed.

Reply:

We thank the Reviewer for this suggestion. We have revised Fig. 4 and Supplementary Fig. 7 for high clarity, in particular for highlighting ordered aromatic ladders between β -strands.

2. These assembly processes are nucleation-dependent processes raising questions of thermodynamic vs kinetic control. This control is hard to address and they nicely approach this selection in their computational analyses and should be highlighted in their claims.

Reply:

We thank the Reviewer for this suggestion. By following the aggregation process from

monomers, to dimers, then to pentamers and multi-stranded β -sheets through simulations, we here demonstrated that the chirality selection within the self-assembly of short amphiphilic peptides was thermodynamically favored. As the Reviewer suggested, we have highlighted this point in the revision.

3. The authors do not address the issue of facial complementarity, so prevalent in cross β assemblies (J. Am. Chem. Soc., 130, 9829-9835), which also may impact nucleation and supramolecular propagation. Why was this issue not considered given its critical role in supramolecular nucleation?

Reply:

We thank the Reviewer for bringing up this point. In fact, we have noticed facial complementarity and realized its importance in cross- β assembly. However, facial complementarity generally occurs between adjacent β -sheets, via their side-chain complementary packing. Driven by this facial complementarity, β -sheets can undergo significant lamination.

In the work recommended by the Reviewer, the facial complementarity between KLVFFAE β -sheets promoted their lamination in the form of Phe zippers, eventually leading to the formation of wide nanotubes (Mehta, A. K. et al. Facial symmetry in protein self-assembly. *J. Am. Chem. Soc.* **130**, 9829–9835 (2008)). We have demonstrated that the facial complementarity between I₃Q GK β -sheets promoted their lateral stacking in the form of polar Gln (Q) zippers, eventually resulting in the formation of wider nanoribbons (Wang, M., et al. Nanoribbons self-assembled from short peptides demonstrate the formation of polar zippers between β -sheets. *Nat.*

Commun. **9**, 5118 (2018)). Despite the critical role of facial complementarity in stabilizing the wide nanotubes and nanoribbons, their relationship to the suprastructure chirality of these higher-order architectures remains elusive. This is partly due to no marked chiral morphologies being observed with these assemblies. On the other hand, Cui et al. have suggested that if lateral adhesion energies were sufficiently large to offset the entropy loss of untwisting β -sheets from their natural state, a flat nanobelt morphology would be expected (Cui, H., et al. Self-assembly of giant peptide nanobelts. *Nano Lett.* **9**, 945–951 (2009)).

In this study, we focused on the side-chain interactions between β -strands within a β -sheet. As for F₃K, we demonstrated that the side-chain aromatic stacking, not only caused the parallel arrangement but also induced the chiral inversion of single β -strands within the β -sheet, therefore controlling the helical handedness of the sheet in the form of aromatic ladders. Although these ordered aromatic ladders between β -strands are somewhat similar to the Phe zipper between β -sheets, we did not observe notable facial complementarity between β -strands in simulations. Given this similarity, we have added a brief discussion about facial complementarity and the recommend paper has been cited in the revised manuscript (Ref. 22 in the revision).

4. A recent publication in this journal (<https://doi.org/10.1038/s41467-024-45019-2>. Editor's Selection) highlights the impact of Phe residues on supramolecular chirality but does not adequately resolve the kinetic vs thermodynamic control or the structural models for assembly, but should be referenced.

Reply:

We thank the Reviewer for recommending this paper. It is very nice work and highly relevant to our present research. We have cited this paper in the revised manuscript (Ref. 10 in the revision).

Reviewer #2:

In this work, the authors investigated the chiral inversion for Phe-based tetrapeptides. It was found the pi-pi stacking of Phe-Phe interactions plays a determinant role in reversing the chirality of the peptides from single-chain strands to multi-chain fibers. This work provides very interesting observations and mechanisms of peptide assembly. The manuscript was well written. It could be further improved with more details about building the structures (pentamers, etc.) and data analyses.

Reply:

We thank the Reviewer for their positive comments. We appreciate their suggestions below, which are very helpful in improving the quality of our manuscript.

Major revisions:

1. What is the “Reduced density gradient”? A brief description is required as well as the reference(s).

Reply:

Done! The Reduced density gradient (RDG), based on the electron density and its first derivative ($s = 1/(2(3\pi^2)^{1/3})|\nabla\rho|/\rho^{4/3}$), is a fundamental dimensionless quantity in DFT (density functional theory) used to describe the deviation from a homogeneous electron distribution. The RDG isosurfaces proposed by John et al. allow us to simultaneously map and analyze a wide range of noncovalent interactions in real space for molecular complexes (John, E. R., Keinan, S., Mori-Sánchez, P., Contreras-García, J., Cohen, A.

J. & Yang, W. Revealing noncovalent interactions. *J. Am. Chem. Soc.* **132**, 6498–6506 (2010)). This reference and relevant description have been included in the revised main text (Ref. 16 in the revision).

2. It seems that Fig. S1(a1-a6) are the same as those in Fig. 1. While protonated Lys was used in Fig. S1b, charge-neutral Lys was provided in Fig. 1b. At neutral solution or pH = 6, Lys is supposed to be fully charged. I see no reason why the charge neutral Lys was provided in the main text.

Reply:

We agree with the Reviewer that Lys is fully charged at neutral solution or pH = 6.0. As a result, we used both protonated and deprotonated peptides in their single strand and dimer simulations. Fig. 1a₁-a₆ (corresponding to Fig. 1a-f of the revised manuscript) and Supplementary Fig. 1a₁-a₆ (corresponding to Supplementary Fig. 1a-f of the revised manuscript) show the conformations of single peptide β -strands where Lys is charge-neutral and positively charged, respectively. It can be seen that for each peptide, the charge-neutral strand exhibited the same twisting direction as the charged strand, despite minor differences in their conformations. RDG analyses also indicated their similar gradient isosurfaces (Fig. 1b₁-b₃ and Supplementary Figs. 1b₁-b₃ and 2. Note that Fig. 1b₁-b₃ and Supplementary Figs. 1b₁-b₃ correspond to Fig. 1g-i and Supplementary Figs. 1g-i of the revised manuscript). Fig. 2 and Supplementary Figs. 3, 4, and 5 show dimer conformations for neutral and positively charged ^LF₃^LK and ^DF₃^LK. Similarly, protonation/deprotonation produced little impact on interstrand interactions and their molecular conformations, especially chiral inversion of single strands within

the parallel dimers.

As indicated in the manuscript, dimers serve as the fundamental structural motif of β -sheets and β -sheet assembly. Once their handedness is determined, the handedness of multi-stranded β -sheets, coming from their stacking along the H-bonding direction, can be well predicted. At the same time, for further quantum chemistry (QC) calculations with larger assemblies such as multi-stranded β -sheets, the charge-neutral state enables rapid and reliable determination of their handedness through a semiempirical QC method (Fig. 5a). Instead, for the charged systems of larger assemblies, an implicit solvent model needs to be added and the calculation workload will be dramatically increased, thus making QC calculations inaccessible at this stage.

To ensure a consistency across length scales, we present simulations of neutral monomers, dimers, and multi-stranded β -sheets in the main text. The simulation results of charged monomers and dimers are provided in Supplementary materials but discussed in the main text.

3. How was the “core-valence bifurcation (CVB) index” defined? A brief description is desired.

Reply:

The core-valence bifurcation (CVB) index is a method to describe the strength of hydrogen bonds, based on the topological analysis of the electron localization function (ELF) (Fuster, F. & Silvi, B. Does the topological approach characterize the hydrogen bond? *Theor. Chem. Acc.* **104**, 13–21 (2000)). The lower the CVB index is, the stronger the H-bond will be. Normal hydrogen bonds can be written as D-H \cdots A, where D is the

hydrogen bond donor atom and A the hydrogen bond acceptor atom. The CVB index is defined as:

$$\text{CVB index} = \text{ELF}(\text{C-V}) - \text{ELF}(\text{DH-A})$$

where ELF(C-V) denotes the bifurcation point value between the core basin and the valence basin, and ELF(DH-A) represents the bifurcation point value between V(D,H) and V(A), that is, the ELF value of the type (3,-1) ELF critical point between H and A atoms.

We have given this brief description in the revised Supplementary materials.

4. (Page 22) “All charged systems were calculated using the solvation model based on density (SMD) solvation model with H₂O as the solvent.” Does it mean that the charge-neutral system was done in a vacuum? How strong are the influences of the implicit hydration models?

Reply:

Yes, QC calculations for the charge-neutral system were performed in vacuum.

As discussed in the Reply to the 2nd major revision from the Reviewer, the results from the simulations for charged peptides using the implicit hydration model were consistent with those from the simulations in vacuum for charge-neutral peptides. Furthermore, for most of neutral molecules, in which the local ionicity is non-obvious in solvent, the implicit solvent model has little effect on their geometric structures and vibration frequency.

5. What is the Kabsch algorithm? Any reference? The pentamer structures were constructed using the Kabsch algorithm and were subjected to DFT optimization.

Please justify that these structures are at an energetically global minimum. Without free energy calculations, the structures are seemingly arbitrary.

Reply:

The Kabsch algorithm is an effective approach for calculating the optimal rotation matrix that minimizes the root mean squared deviation (RMSD) between two paired sets of points (backbone H-bonding donors and acceptors in our study), thereby translocating and rotating molecules for the best structure alignment within their assemblies. (Ref. 21 in the revision: Kabsch, W. A solution for the best rotation to relate two sets of vectors. *Acta Crystallogr., Sect. A: Cryst. Phys. Diffr., Theor. Gen. Crystallogr.* **32**, 922–923 (1976)) and https://en.wikipedia.org/wiki/Kabsch_algorithm).

After molecular packing and aligning via the Kabsch algorithm, pentamers were subjected to structural optimization using DFT calculations, giving rise to the structures with minimum energies. Hence, the pentamer structures shown in Fig. 4b and Supplementary Fig. 7b are not arbitrary.

6. I am a bit surprised that no micellar structures were formed from these peptides. Atomistic MD simulations are desired to justify the fiber morphology, instead of micelles.

Reply:

Amphiphilic peptides differ from conventional surfactants in their backbones being composed of a string of amide bonds (-CH(R)-CO-NH-). Driven by backbone-backbone hydrogen bonding, amphiphilic short peptides usually adopt β -sheet conformations and form one dimensional (1D) nanostructures such as nanofibrils,

nanoribbons, and nanotubes, distinct from conventional surfactants that readily aggregate into micelles. Paramonov et al. have confirmed that backbone-backbone hydrogen bonding played a crucial role in the self-assembly of peptide amphiphiles into nanofibrils (Paramonov, S. E., et al. *J. Am. Chem. Soc.* **128**, 7291–7298 (2006)). They found that after blocking hydrogen bonds via N-methylation, these amphiphiles formed spherical micelles rather than long nanofibrils. Isoleucine (I) and leucine (L) are isomeric amino acids with different side chain branching modes: isoleucine shows a stronger propensity for β -sheet structuring than leucine in protein folding. As a result, we have demonstrated that I₃K adopted β -sheet secondary structures and self-assembled long nanofibrils while L₃K adopted random coil conformations and formed spherical stacks (Han, S., et al. *Chem. Eur. J.* **17**, 13095–13102 (2011)).

At the same time, we have used replica exchange molecular dynamics (REMD) to evaluate the initial aggregation behaviors of I₃K and L₃K (Zhou, P., et al. *Langmuir* **32**, 4662–4672 (2016)). Our simulation results indicated that I₃K packed into dimer and trimer aggregates along the H-bonding direction while L₃K tended to form dimer and trimer aggregates without intermolecular H-bonds, consistent with our experimental results. These I₃K oligomeric aggregates, acting as nucleation points, direct subsequent packing of I₃K molecules along the H-bonding direction, resulting in β -sheets, followed by lateral stacking of β -sheets into fibers.

Additionally, because Phe has similar hydrophobicity and β -sheet forming propensity to Ile, F₃K also adopts β -sheet conformations and forms long nanofibrils, as discussed in our manuscript.

Minor revisions:

1. In the MD simulations and QM calculations, how were the C- and N-terminals treated?

Both are seemingly charge neutral in Fig. 1. Please confirm.

Reply:

Yes, the two termini are charge-neutral, as a result of C-terminal amidation and N-terminal acetylation. Such a treatment is to avoid the interference of additional terminal electrostatic interactions on our simulations. In fact, the peptides used in our experiments were also terminally capped during their synthesis.

2. (Pag 5-6) $^{La}I_3^L K$ & $^{Da}I_3^D K$. Replace La with L, and Da with D.

Reply:

Isoleucine has two chiral centers, i.e., the α -carbon (C_α) and the side-chain β -carbon (C_β) atoms. Correspondingly, there are two pairs of isoleucine enantiomers, i.e., L-isoleucine ($2S, 3S$) and D-isoleucine ($2R, 3R$), and L-*allo*-isoleucine ($2S, 3R$) and D-*allo*-isoleucine ($2R, 3S$). Thus, in the two peptides $^{La}I_3^L K$ and $^{Da}I_3^D K$ in our manuscript, isoleucine residues denote L-*allo*-isoleucine and D-*allo*-isoleucine, respectively. These notations must be kept distinguishing them from each other.

3. (Page 7) “brown patch in the enlarged area of Fig. 1b1”. I suppose the authors are referring to the “green patch”. Please clarify and be specific.

Reply:

We thank the Reviewer for their careful review of this manuscript. In fact, the blue, green, and red regions within the RDG isosurfaces represent strong attractive H-

bonding, weak van der Waals interactions, and strong steric repulsion, respectively, as illustrated by the following figure (John, E. R., Keinan, S., Mori-Sánchez, P., Contreras-García, J., Cohen, A. J., Yang, W. Revealing noncovalent interactions. *J. Am. Chem. Soc.* **132**, 6498–6506 (2010); Lu, T. & Chen, F. Multiwfn: A multifunctional wavefunction analyzer. *J. Comput. Chem.* **33**, 580–592 (2012)).

In this manuscript, the brown patch shown in Fig. 1b₁ (corresponding to Fig. 1g of the revised manuscript) indicates a strong steric hindrance between the benzene ring of Phe3 and the local backbone of F₃K. We have clarified this point in the revision.

4. Please provide the full name of “QC”.

Reply:

We have provided the full name of QC, i.e., quantum chemistry in the revision.

Reviewer #3:

This manuscript uses a previously developed theoretical method combining quantum chemistry calculations and molecular dynamics simulations in order to reliably determine the single Beta-strand conformations and chirality based on intrastrand non-bonded interactions. Their computational results are compared with synthesis and atomic force microscopy results. By providing an effective approach for unraveling

how hydrophobic interactions control the handedness evolution of peptide beta-sheet assemblies, the study opens an avenue to harness peptide aggregates with targeted handedness via aromatic side chain interactions.

Reply:

We thank the Reviewer for their positive comments.

This investigation is of interest to a wide audience and appears to be original. The methods used are satisfactory, although I would be curious if qualitative results hold with different functionals.

Reply:

We appreciate the positive assessment of the originality of our manuscript from the Reviewer.

As for other functionals, the use of mainstream functionals such as M06-2X and PWPB95 can also lead to the same qualitative results. However, for some earlier proposed functionals such as BLYP and B3LYP, if the DFT-D3 dispersion correction was not incorporated, wrong qualitative results were typically obtained. This is because these old functionals cannot describe dispersion interactions well, which virtually contribute significantly to peptide self-assembly. As a result, the DFT-D3 dispersion correction is required in order to reliably describe the weak interactions when using these functionals.

I do think the presentation could be improved to make the presentation and discussion of their results clearer. As it stands now it is work for the reader to figure out when theoretical predictions are being made vs. when they are explicitly being compared to experiment. Perhaps a table when discussing the Kabasch algorithm predictions. With

a little bit of clarification and discussion of chosen methods I would recommend this paper for publication.

Reply:

We agree with the Reviewer that the manuscript can be improved by making the chosen methods and discussions much clearer. Correspondingly, we have included more descriptions and discussion of these methods such as the CVB index and the Kabsch algorithm in the revised manuscript, as indicated above.

We appreciate the valuable comments and suggestions from the three Reviewers that are extremely helpful in improving the quality of the manuscript and providing ideas for future work.

REVIEWERS' COMMENTS

Reviewer #1 (Remarks to the Author):

The author's comments and corrections greatly improve the quality of the presentation of the science and I approve publication.

Reviewer #2 (Remarks to the Author):

My comments have been appropriately addressed. I recommend its publication as it.